# A Cell-Autonomous Oncosuppressive Role of Human RNASET2 Affecting ECM-Mediated Oncogenic Signaling

**DOI:** 10.3390/cancers11020255

**Published:** 2019-02-22

**Authors:** Francesca Roggiani, Cristina Riva, Francesco Raspagliesi, Giovanni Porta, Roberto Valli, Roberto Taramelli, Francesco Acquati, Delia Mezzanzanica, Antonella Tomassetti

**Affiliations:** 1Unit of Molecular Therapies, Department of Research, Fondazione IRCCS Istituto Nazionale dei Tumori, Via Amadeo 42, 20133 Milan, Italy; francesca.roggiani@istitutotumori.mi.it (F.R.); delia.mezzanzanica@istitutotumori.mi.it (D.M.); 2Unit of Pathology, Department of Medicine and Surgery, Università degli Studi dell′Insubria, via JH Dunant 5, 21100 Varese, Italy; cristina.riva@uninsubria.it; 3Gynecology Oncology Unit, Department of Surgery, Fondazione IRCCS Istituto Nazionale dei Tumori, Via Amadeo 42, 20133 Milan, Italy; Francesco.raspagliesi@istitutotumori.mi.it; 4Medical and Human Genetics unit, Department of Medicine and Surgery, Università degli Studi dell′Insubria, via JH Dunant 5, 21100 Varese, Italy; giovanni.porta@uninsubria.it (G.P.); roberto.valli@uninsubria.it (R.V.); 5Department of Biotechnology and Life Sciences, Università degli Studi dell′Insubria, via JH Dunant 5, 21100 Varese, Italy; Roberto.Taramelli@uninsubria.it

**Keywords:** RNASET2, epithelial ovarian cancer, extracellular matrix, src kinase, src family kinases

## Abstract

RNASET2 is an extracellular ribonuclease endowed with a marked antitumorigenic role in several carcinomas, independent from its catalytic activity. Besides its antitumorigenic role by the recruitment to the tumor mass of immune cells from the monocyte/macrophage lineage, RNASET2 is induced by cellular stress and involved in actin cytoskeleton remodeling affecting cell interactions with the extracellular matrix (ECM). Here, we aimed to investigate the effects of RNASET2 expression modulation on cell phenotype and behavior in epithelial ovarian cancer (EOC) cellular models. In silico analysis on two publicly available datasets of gene expression from EOC patients (*n* = 392) indicated that increased RNASET2 transcript levels are associated with longer overall survival. In EOC biopsies (*n* = 101), analyzed by immunohistochemistry, RNASET2 was found heterogeneously expressed among tumors with different clinical–pathological characteristics and, in some cases, its expression localized to tumor-associated ECM. By characterizing in vitro two models of EOC cells in which RNASET2 was silenced or overexpressed, we report that RNASET2 expression negatively affects growth capability by conferring a peculiar cell phenotype upon the interaction of EOC cells with the ECM, resulting in decreased src activation. Altogether, these data suggest that drugs targeting activated src might represent a therapeutic approach for RNASET2-expressing EOCs.

## 1. Introduction

Epithelial ovarian cancer (EOC) is the fifth leading cause of cancer-related deaths among women, and the leading cause of death from gynecological cancer. EOCs incidence and mortality rates have remained unchanged over the last 30 years due to intrinsic or acquired chemoresistance (for review see [1]). The most recent classification of EOCs identifies type I tumors that tend to be low-grade, genetically stable neoplasms, and type II tumors that represent high-grade EOCs characterized by high chromosomal instability and intratumor heterogeneity [2], features that likely govern the clinical end-points, including the acquisition of metastatic potential and chemotherapeutic resistance. High grade EOCs are frequently diagnosed when the patients present peritoneal solid primary and secondary lesions associated, in ~40% of the patients, to malignant ascites rich of tumor multicellular aggregates (MCAs). These MCAs overcome anoikis, persist as ascites, and attach on the abdominal peritoneum or omentum, suggesting that the attachment through integrins of cancer cells onto the mesothelial cells covering the basement membrane is an important key step in EOC dissemination [3].

Human RNASET2, a ribonuclease that belongs to the T2 family, has recently emerged as an appealing new player in oncology displaying marked tumorigenic and metastatic suppressor properties. The RNASET2 protein is usually detected in three isoforms: a ′full length′ form of 36 KDa and two other isoforms, of 27 KDa and 31 KDa, produced by proteolytic cleavage at the C-terminal of the full length form during transport to the secretory pathway [4]. The protein is in fact found in the cytoplasm at the periphery of the perinuclear region, where it colocalizes with markers of the endoplasmic reticulum and both cis/trans Golgi [5]. Strikingly, RNASET2 exerts its ‘antitumor function’ independently from its ribonuclease activity and we have demonstrated that its expression in tumor cells in vivo is associated with the recruitment into the tumor mass of cells from the monocyte/macrophage lineage committed to anticancer activity [6,7,8,9,10]. In keeping with these findings, RNASET2 transcript levels were found to be downregulated in EOC sample and cell lines compared to their normal counterpart [6].

Recently, a new mode of action of this protein was assessed that led to the hypothesis that RNASET2 may affect cancer growth by affecting the ECM/integrin signaling pathway as well. In fact, silencing of RNASET2 expression in the high grade serous EOC cell line OVCAR3 triggered a marked disruption of the network of actin filaments and stress fibers, inducing a pattern of peripheral actin filaments [9], thus leading to increased migration and adhesion to laminin, collagen I, and IV. Of note, the adhesion of these cells on collagen I was accompanied to an increase of paxillin activation and, accordingly, to an increase of mature focal adhesions (FAs) [9]. In addition, soluble recombinant RNASET2 protein delivered to RNASET2-silenced cells was able to reorganize the actin cytoskeleton in these cells [11].

The present study aimed to investigate the clinical relevance of RNASET2 expression in EOCs and to further clarify its involvement in controlling the growth properties of EOC cells in the intent to look for a therapeutic strategy specific for the ‘RNASET2 phenotype’.

## 2. Results

### 2.1. RNASET2 Transcript Expression in EOC Is Associated to Better Prognosis

In keeping with experimental data suggesting its role as a tumor suppressor [7,12,13], RNASET2 expression has been so far associated to less aggressive phenotypes in several cancer models, including ovarian cancer [7,8,9,13,14]. To investigate the clinical relevance of RNASET2 expression in EOCs, the correlation of RNASET2 expression and overall survival (OS) was analyzed in two publicly available datasets—GSE26193 [15] and GSE9891 [16]—containing gene expression data from 392 EOC patients. In both datasets, patient with low expression of RNASET2 transcript experienced a shorter OS as compared to patients with higher RNASET2 transcript expression (Figure 1a, upper panel), as shown by the Kaplan–Meier plots (Figure 1a, lower panels) (log-rank test, *p* = 0.023; HR = 1.89 (CI 1.1–3.3), and *p* = 0.0075, HR = 1.82 (CI 1.2–2.28), respectively). 

We then performed an immunohistochemistry (IHC) analysis in a case material of 101 EOC samples, representative of EOC different histotypes and grades, to evaluate RNASET2 protein expression and localization. Although 73% of EOC samples showed RNASET2 protein expression (Table 1), only in 32% of them were at score 2, with no association to a particular EOC subtype, and basically recapitulating the data observed for the relevant transcript of panel a.

Besides the staining intensity, the RNASET2 signal was homogeneously found at the cytoplasm level or diffusely present at the level of ECM deposition, likely due to protein secretion by cancer cells (representative images in Figure 1b). Although at different intensities (see Table 1), RNASET2 expression was also detected both in normal ovarian and tubal epithelia (Figure 1b, upper panels), from which different histotypes of EOC can arise [17]. Follow-up data were not available for this cohort of patients, thus preventing the possibility to associate RNASET2 protein expression to patients’ prognosis.

In agreement with the proposed oncosuppressive role of RNASET2, these data indicate that high levels of RNASET2 transcript levels are associated to better prognosis for EOC patients. In addition, RNASET2 protein can be found accumulated in the cytoplasm or in tumor-associated ECM.

### 2.2. RNASET2 Depletion Causes Phenotypic Changes in EOC Cellular Models

In order to investigate the role of RNASET2 in EOC cells expressing different levels of the protein, two in vitro EOC models were established. The RNASET2-expressing OAW42 EOC cell line, displaying an epithelial morphology [18,19], was stably silenced for RNASET2 expression by RNA interference. By contrast, the RNASET2-expression negative SKOV3 EOC cell line, with a spindle-like morphology [18,19,20], was chosen for stable transfection with RNASET2 expression vectors. Both transfectants were then biochemically and functionally characterized.

After depletion of RNASET2, OAW42 cells acquired dramatic changes in the actin cytoskeleton with loss of the membrane actin ring typical of epithelial cells and appearance of ticker stress fibers, stained with fluorescent phalloidin, with loss of cell–cell contacts, as shown by immunofluorescence (IF) assays (Appendix A, upper panels).

Untransfected SKOV3 cells showed barbed ends of actin filaments, suggestive of lamellipodia of migrating cells, while RNASET2-transfected SKOV3 cell lost these structures, although preserved stress fibers (Appendix A, lower panels). Of note, treatment with the human recombinant RNASET2 was able to revert the cytoskeleton assembly of RNASET2-silenced OAW42 cells. Conversely, the same treatment on RNASET2 not expressing parental SKOV3 cells caused a shift from a mesenchymal phenotype to a more rounded epithelial-like shape, with fewer protrusions and increased cell–cell contacts (Appendix A, upper and lower right panels, respectively).

Since both transfectants showed morphological differences upon modulation of RNASET2 expression (Appendix A), we first intended to analyze whether OAW42 and SKOV3 cells could have undergone RNASET2-mediated epithelial–mesenchymal transition (EMT) or the reverse process, respectively. Real-time RT-PCR confirmed the knockdown of RNASET2 transcript upon transfection of the shRNASET2 construct in OAW42. Moreover, whereas no differences in the expression of the E-cadherin-encoding CDH1 gene was observed, a 2-fold increase of the N-cadherin-encoding CDH2 gene was observed upon RNASET2 silencing (Figure 2a, upper panels). We did not observe significant variation in the expression levels of ZEB1 and SNAI1—two genes encoding E-boxes transcription factors responsible for inhibition of CDH1 transcription [21] (Appendix A, upper panels). The transcripts for ZEB2 and SNAI2 were not detectable in both cell lines and the correspondent transfectants [22].

When the same analysis was performed on SKOV3 cells upon stable transfection of the RNASET2 cDNA, the RNASET2 transcript was found to be increased ~9 times compared to mock-transfected cells and associated to an increase of both CDH1 and CDH2 transcript levels (Figure 2a, lower panels) with a corresponding slight decrease of ZEB1 and SNAI1 (Appendix A, lower panels). 

At the protein level, RNASET2 isoforms were not detected in shRNASET2-OAW42 lysates by Western blot (WB) analysis, as expected. In these cells, N-cadherin and the epithelial marker cytokeratin 8/18 increased ~3 times, while the mesenchymal protein vimentin increased ~2 times by WB analysis (Figure 2b). The levels of E-cadherin transcript and protein were apparently unchanged or slightly increased (Figure 2a,b, respectively) as confirmed by the junctional E-cadherin pattern observed by IF (Figure 2c); the increase in N-cadherin expression was also confirmed by IF analysis, in which the protein was detected at the sites of cell–cell contacts in shRNASET2-OAW42 cells (Figure 2b,c). 

It is interesting to note that, upon RNASET2 overexpression in SKOV3 cells, no marked changes in either cytokeratin or vimentin expressions were observed by WB (Figure 2b). Thus, besides the observed increase of N-cadherin in both cell lines following modulation of RNASET2 expression, the transfectants appeared to undergo neither the EMT nor the reverse process. Altogether, these data indicate that the morphological changes observed following RNASET2 knockdown or overexpression in both EOC cells was likely aroused by actin cytoskeleton remodeling. 

### 2.3. RNASET2 Expression Negatively Impinges EOC Cell Growth

We then asked whether certain cancer-related parameters were affected in vitro following RNASET2 modulation in both cellular models. In vitro proliferation assay clearly demonstrated a statistically significant increase in the growth potential of RNASET2-silenced OAW42 cells (shRNASET2-OAW42, Figure 3a, left panel). Accordingly to these results, overexpression of RNASET2 in SKOV3 cells decreased their growth capability ~2.5-fold (Figure 3a, right panel). In keeping with these data, anchorage-independent growth assays indicated that RNASET2 expression was associated to a significant lower capability to form colonies in soft agar in both cell lines (Figure 3b and Appendix A). By showing the in vitro oncosuppressive role of RNASET2 in two EOC cell lines endowed with different aggressiveness, these data strongly support the previously reported in vivo oncosuppressive role of RNASET2 [8,14].

Cell growth potential was also assessed in both cell lines grown as 3-dimentional cultures (3D) in Algimatrix^TM^ (ThermoFisher Scientific, Waltham, MA, USA), in which cells grow independently from integrin adhesion, similarly to the MCAs present in EOC ascites [23]. Surprisingly, shRNASET2-OAW42 spheroids formed in Algimatrix^TM^ were ~2-fold less than those obtained from mock-OAW42 cells (Appendix A, left panel), not confirming the previous results but arguing for the hypothesis that integrins can be causally involved in RNASET2-dependent growth signaling.

By contrast, RNASET2-transfected SKOV3 cells formed a lower number of spheroids in Algimatrix^TM^ (right panel) confirming an inhibitory effect upon RNASET2 expression in these EOC cells.

To get further insight on the RNASET2 role in growth/migration, transfectants of both OAW42 and SKOV3 EOC cell models were also seeded onto Matrigel^®^, as described in the Materials and Methods section, for evaluating the behavior of the cells in contact with ECM-related scaffold while proliferating. After seven days, both mock- and RNASET2-silenced OAW42 cells formed visible spheroids of different sizes in the Matrigel^®^, whereas no cell monolayer was detected. Of note, shRNASET2-OAW42 cells displayed bigger spheroids (Figure 3c). Altogether, these observations indicate the tendency of OAW42 cells to aggregate instead of migrate, probably due to their epithelial-like phenotype and the presence of E-cadherin on their cell membrane (Figure 2b,c). 

On the other hand, mock- and RNASET2-transfected SKOV3 showed a completely different behavior, since mock-transfected cells adhered and grew onto Matrigel^®^, whereas RNASET2-overexpressing SKOV3 cells grew as spheroids (Figure 3c), suggestive of an epithelial-like phenotype and the tendency to increase cell–cell contacts.

These data support the hypothesis that RNASET2 expression confers to EOC cells an ECM-dependent ‘RNASET2 phenotype’ characterized by decreased growth capability, both in 2D and 3D experimental conditions, further supporting our previously observations that RNASET2 can negatively affect adhesion to ECM proteins and cytoskeletal remodeling [9]. 

### 2.4. ‘RNASET2 Phenotype’ Negatively Modulates Src Activation

In order to assess the molecular mechanism elicited by the interaction between RNASET2-expressing cells and ECM, we decided to define the role of the ECM proteins to which each cellular model could bind, according to their expression pattern of specific membrane β integrin subunits. 

Thus, we first analyzed integrin expression in OAW42 cells by flow cytometry. Although OAW42 cells express both β1 and β3 integrins (Appendix A, upper panel), they are able to adhere to fibronectin but not to collagen (Appendix A). SKOV3 cells were instead tested on collagen since they showed β3 integrin expression (Appendix A). When expressed, the 36 KDa RNASET2 isoform was also found secreted in the conditioned medium of both OAW42 and SKOV3 cells (Appendix A), and growth of these cells on the relevant ECM protein seemed not to affect the levels of the secreted protein. According to the activation induced by tumor cells/ECM interaction [24], src phsphorylation was therefore analyzed upon adhesion to fibronectin or collagen I for starved OAW42 and SKOV3, respectively. Strikingly, the presence of the ‘RNASET2 phenotype’ clearly inhibited src activation in both cell lines (Figure 4a,b) and, upon adhesion on the specific ECM protein (fibronectin or collagen I), both AKT and MAPK phosphorylation levels increased in the absence of RNASET2 expression and were both inhibited by the src family kinase (SFK) inhibitor PP2, indicating that both signalings were SFK-dependent. Accordingly, staining with antiphosphorylated paxillin evidenced fully mature FAs colocalizing with actin only in RNASET2 not-expressing EOC cells (Figure 4c).

So far, these evidences strongly indicate that RNASET2 expression in both EOC cells is associated with a less aggressive tumor phenotype in terms of proliferation, with inhibition of ECM-dependent src kinase activation.

### 2.5. Src Inhibitor Restores ‘RNASET2 Phenotype’

To investigate whether src activation was responsible for the growth behavior of the cells interacting with ECM, as reported in Figure 3c, cell adhesion following growth of both transfectants was assessed in the presence of the SFK inhibitor PP2 in Matrigel^®^. Noteworthy, RNASET2-depleted OAW42 treated with PP2 displayed small spheroids in Matrigel^®^, comparable to mock-transfected cells (Figure 5, left panels). By contrast, the PP2 effect on growth in Matrigel^®^ was more evident in mock-transfected SKOV3 (right panel), and spheroids were comparable to those displayed by RNASET2-transfected SKOV3. Furthermore, the latest cells appeared smaller, indicating a cytostatic effect played by the src inhibitor.

These observations further support the notion that SFK activation is responsible for the phenotype and the growth advantage of EOC cells not expressing RNASET2.

### 2.6. EOC Cells with ‘RNASET2 Phenotype’ Are Sensitive to Src Inhibitor

Finally, we decided to test the efficacy of PP2 in these two EOC models. As expected, cells expressing RNASET2 were more sensitive to the cytostatic effect of PP2. In particular, RNASET2-silenced OAW42 cells showed an IC50 2-fold higher than mock-transfected cells (Figure 6). Conversely, RNASET2-overexpressing SKOV3 cells displayed an IC50 3 times lower than control SKOV3 cells not expressing RNASET2.

Altogether, these findings suggest that inhibition of SFK activation might represent a novel therapeutic approach to be tested for RNASET2-expressing EOCs, including more aggressive tumors.

## 3. Discussion

In this manuscript, we have defined the oncoppressor RNASET2 gene as a new player in the control of the aberrant interaction between cancer cells and ECM. In particular, our in vitro data indicate that expression of RNASET2 in EOC cells confers an ECM-related phenotype characterized by decreased growth capability and marked actin cytoskeleton remodeling, mainly due to decreased src kinase activation, with consequent increased susceptibility of these cells to SFK inhibition. 

Several research groups including ours have previously reported the antitumor effect of RNASET2 on both in vivo and in vitro experimental models [12,25,26,27]. Accordingly, the recombinant human RNASET2 has been proposed as anticancer drug [28]. Of particular relevance, RNASET2 was shown by our group to signal to the innate immune system in in vivo xenograft-based models of human ovarian cancer by attracting host macrophages to the tumor mass and possibly contributing to their M1 polarization [10,14]. In addition to this noncell-autonomous oncosuppressive role, we and others have recently demonstrated that RNASET2 behaves as an oncosuppressor gene in a cell-autonomous way as well, by acting as a stress response gene whose expression and secretion increases under several stressful conditions, including hypoxia [9]. The present findings represent a step forward to define the role of RNASET2 in the progression of tumors, in particular in EOCs. In fact, the known behavior of *RNASET2* as a stress response gene [9,26] led us and other authors to propose a role for the RNASET2 protein as an alarmin-like molecule [14,29].

Given our data of RNASET2 protein heterogeneously expressed in a panel of EOC biopsies with no particular correlation with pathological characteristics and those obtained exploiting RNASET2 expression profiles in publicly available datasets, further efforts will be devoted to investigate whether RNASET2 protein could be used as prognostic marker for EOC patients.

We show here that RNASET2 staining could be also detected in tumor-associated ECM, pointing at the tumor cell–ECM interaction as a plausible novel biological process involved in RNASET2-mediated tumor suppression. This observation together with our previous finding of an RNASET2-mediated inhibition of adhesion to ECM proteins, such as laminin, collagen I, and IV, obtained by silencing of RNASET2 expression in the high grade serous EOC cell line OVCAR3 [9], prompted us to analyze more in detail the in vitro biological effect of RNASET2 expression in two EOC cellular models. Based on both their in vitro molecular characteristics and in vivo functional behavior when xenotransplanted in nude mice [18,19,20], we have chosen OAW42 and SKOV3 as poorly aggressive/RNASET2-expressing and aggressive/RNASET2-negative EOC cell lines, respectively, as experimental models to study the biochemical and functional changes associated to what we called the ‘RNASET2 phenotype’. 

By experimentally manipulating RNASET2 expression levels in both cell lines, the less aggressive behavior observed in RNASET2-expressing cells, evaluated here as both proliferation of viable cells in 2D culture and adhesion and growth in Matrigel^®^ as 3D culture, was apparently not attributable to a clearly defined EMT-associated process (such as reduction of E-boxes transcription factors and the cadherin switch) but rather to stabilization of the epithelial characteristics, with increased E-cadherin cell–cell contacts and markedly changes in the actin cytoskeleton. As far as the RNASET2 effects on the actin cytoskeleton is concerned, these results are not surprising since other members of the T2 RNase protein family, like the trematode Omega-1 [30] and fungal ACTIBIND orthologs [31], have been reported to induce marked rearrangements of the actin cytoskeleton organization. Indeed, we also have previously shown that silencing of RNASET2 in OVCAR3 cells led to cytoskeletal reorganization, with a consequent change in the cellular phenotype, from mesenchymal to amoeboid [9]. In some studies, this ability to reorganize the actin cytoskeleton has been attributed by some authors to the ability of ACTIBIND and RNASET2 to directly bind F-actin in vitro [31], thus suggesting a direct role of these proteins on this cytoskeletal component. Despite these results, we found no colocalization of RNASET2 with the actin cytoskeleton by confocal microscopy analysis [9] and both immunoprecipitation assays and further analysis using confocal microscopy on OVCAR3 cells clearly showed that RNASET2 is not able to form complexes with actin [22], thus suggesting an indirect action of the protein on the cytoskeleton. Indeed, our present in vitro data indicate that RNASET2 could act as a sensor able to thwart protumorigenic signals exerted by the interactions between tumor cells and ECM. 

In this context, it is worth noting that a marked change in the actin cytoskeleton was observed not only by manipulating the endogenous expression level of RNASET2 in our cellular models, but also upon cell treatment with soluble recombinant RNASET2 protein. This observation suggests a main role exerted by the secreted, extracellular isoform of this protein in cell-autonomous tumor suppression and at the same time unveils the possible occurrence of a paracrine effect of RNASET2, possibly acting on a heterogeneous cancer cell population endowed with different levels of RNASET2 expression and secretion. We reckon that the observed immunostaining for RNASET2 protein expression at the site of ECM on our EOC biopsies panel might be considered an in vivo confirmation of this hypothesis. 

A clear cut-off of RNASET2 expression for phenotype definition is not yet available; however, our data indicate that the expression of RNASET2 detected in SKOV3 is not sufficient to exert an antitumor activity which is obtained only after protein over-expression.

Cancer onset and progression are widely attributed to the aberrant interaction between cancer cells and the microenvironment. In fact, ECM deposition around primary tumors and in the metastatic niche is known to mainly contribute to the activation of ECM/integrin-dependent pathways that are responsible for several biophysical, biochemical and functional features of cancer cells, such as proliferation, migration, and invasion [24]. Importantly, this ECM/integrin crosstalk triggers src activation in several tumors, including EOCs [32,33]. Furthermore, we previously showed that collagen organization profoundly changed upon de novo expression of RNASET2 in tumors arised from an EOC cell line xenotransplanted in nude mice [7,8,9,13,14]. In this study, RNASET2 expression appears to negatively affect the interaction of tumor cells with ECM proteins in vitro in two independent EOC cellular models characterized by a marked difference in their aggressiveness, pointing to a widespread oncosuppressive role for RNASET2 in EOC. Of note, the observed oncosuppressive role of RNASET2 was likely acting at the cellular mechanotransduction level, which takes the form of a modification of the actin cytoskeleton and decreased src activation. Although no information about src activation in tubal and ovarian epithelia is reported, it is well-known that the epithelial E-cadherin-mediated cell–cell junctions are disrupted by src activation [34,35], so we can reasonably infer that the src pathway is not active in those epithelia. By contrast, src has consistently been found to be expressed and activated in both type I and II EOC patient samples [33] and its targeting caused the inhibition of both tumor and vascular networks in a murine model of EOC [36]. No information about differential activation of src or the other SFKs in EOCs with different histotype are reported. On the other hand, several EOC cell lines resulted sensitive to another SFK inhibitor, the dasatinib [37], thus suggesting possible therapeutic combinations. However, data reported after clinical trials with SFK inhibitors—dasatinib or saracatinib (for review see [38])—have shown a poor response rate and no biomarkers of response were identified indicating that a better selection of EOC patients is required.

Strikingly, we show here that src activation is negatively affected by RNASET2 expression and the use of the SFK inhibitor in RNASET2 nonexpressing cells restores the ‘RNASET2 phenotype’ in these cells, strongly suggesting SFK targeting as a therapeutic approach for RNASET2-expressing EOCs. Experiments are ongoing to test the efficacy of SFK inhibitor dasatinib in preclinical models of EOC patient-derived xenograft [39] expressing different levels of RNASET2.

Furthermore, it is worth remembering that VEGF inhibition by Bevacizumab has entered in the first-line therapy as standard drug together with platinum-based chemotherapy in EOC patients [40] and since src is crucial to preserve the integrity of the endothelium [41], the inhibition of SFK in combination with antiangiogenic therapies could significantly implement the therapeutic efficacy on EOC cells. Therefore, further preclinical and clinical studies are needed to better validate SFK targeting in EOC subgroups according to their ‘RNASET2 phenotype’.

## 4. Materials and Methods 

### 4.1. Cell Lines and Plasmids

The EOC cell lines used in this study were OAW42, kindly provided by Dr Ulrich (Dr. A Ullrich, Martinsried, Germany), and SKOV3; both were purchased from ATCC. OAW42 cell clones were stably transfected with control scrambled shRNA or RNASET2-targeting shRNAS, as described for OVCAR3 cells in [9]. SKOV3 clones were stably transfected with a pcDNA3-based expression vector encoding RNASET2 and eGFP or eGFP alone, for control cells. Transfection was performed using Lipofectamine 2000 from ThermoFisher Scientific (Waltham, MA, USA) according to the manufacturer’s protocol. Cells were maintained at 37 °C in a humidified atmosphere of 5% CO_2_ in EMEM (Sigma Aldrich, St. Louis, MO, USA) supplemented with 10% FCS (Gibco, ThermoFisher Scientific), 2 mM L-glutamine (Lonza, Basel, Switzerland), 1% AANE (Lonza), and 2 μg/mL puromycin (Gibco, ThermoFisher Scientific) for mock and shRNASET2-OAW42; RPMI 1640 medium supplemented with 10% FCS, 2 mM L-glutamine 400 μg/mL G418 (Fisher BioReagents, PA, USA) for mock, and tRNASET2-SKOV3-GFP. Cells were genotyped at the functional genomic facility of our institute using a Stem Elite ID System (Promega, Madison, WI, USA), according to the manufacturer’s instructions and ATCC guidelines. Cells were routinely confirmed to be mycoplasma-free by a MycoAlert Mycoplasma Detection Kit (Lonza, Basel, Switzerland).

### 4.2. Patient Samples

We analyzed by IHC with anti-RNASET2 Ab with a total of 101 FFPE histological selections from EOC patients. Seventy-two were primitive samples collected by Service of Pathological Anatomy and Division of Obstetrics and Gynecology of the Hospital Circolo Fondazione Macchi of Varese (Italy), from 1994 to 2012. Twenty-nine were collected at Fondazione IRCCS Istituto Nazionale Tumori. The Institutional Review Board approved the use of archived material, as well as clinical data. All clinical specimens were accompanied by informed consent from all patients to use the excess biological material for investigative purposes.

### 4.3. In Silico Analysis of RNASET2 Expression and Survival Analysis

Gene expression datasets publicly available on GEO repository (http://www.ncbi.nlm.nih.gov/geo/) were considered. The log2 expression of RNASET2 identified by 217984_at Affymetrix probe set was retrieved from GSE26193 [15] and GSE9891 [16] datasets. Kaplan–Meier Plotter (http://kmplot.com) was exploited for survival analyses [42].

### 4.4. IHC

IHC with anti-RNASET2 Ab was performed on the human EOC samples reported above. The staining of paraffin-embedded formalin-fixed tissues was performed essentially as described [43]. Antigen retrieval was carried out at 96 °C for 5 min in citrate buffer (pH 6) and incubation with anti-RNASET2 antibody (5 μg/mL) was carried out overnight at 4 °C. Positivity or negativity of staining was assessed independently by two observers (C.R. and A.T.). The score was assigned as follows; score 0 for a negative staining of tumor cells, score 1 for moderately stained tumor cells, and a score 2 for strong stained tumor cells.

### 4.5. Antibodies

The list of the primary and secondary Abs, as well as working dilution for each assay, is reported in Appendix A.

### 4.6. RNA Extraction and Quantitative Real-Time RT–PCR Analysis

The kit isolation of small and large RNA and NucleoSpin^®^ miRNA (Macherey-Nagel, Düren, Germany) was utilized for total RNA extraction, while TaqMan^®^ Gene Expression Assays (Applied Biosystems, Foster City, CA, USA) were used for quantitative real-time RT-PCR, according to the manufacturers’ protocols.

### 4.7. Western Blotting

Cells were washed with ice-cold PBS containing Na_3_VO_4_ 0.1 mM and lysed with NuPAGE^®^ LDS sample buffer (1×) (ThermoFisher Scientific) under reducing conditions, as previously described [23]. Samples were loaded on precast Bolt Bis-Tris Plus Gels (ThermoFisher Scientific) and blotted in a dry system. Quantization on WB was assessed by using ImageJ software. The secreted RNASET2 isoform was analyzed in 24-h conditioned media without FBS of confluent RNASET2-expressing OAW42 and SKOV3 cells grown on plastic, fibronectin, or collagen I in a 24-well plate.

### 4.8. Immunofluorescence (IF) and Confocal Microscopy

Cells were grown adherent on 8-well glass chamber slides (Nalge Nunc International, New York, NY, USA). The IF was performed as described [23]. For analysis of phosphorylated paxillin, cells were analyzed after adhesion on fibronectin or collagen I. Samples were analyzed using an Eclipse TE2000-S microscope with a 40× 0.75NA PanFluor objective (Nikon, Tokyo, Japan). Images were acquired with ACT-1 software (Nikon) and processed using ImageJ and Adobe Photoshop softwares. Confocal microscopy was carried out using a Leica TCS SP8 X confocal laser scanning microscope (Leica Microsystems GmbH, Mannheim, Germany). Images were acquired in the scan format 1024 × 1024 pixels in a single plane using a HC PL APO CS2 60×/1.30 oil-immersion objective and a pinhole always set to 1 Airy unit and analyzed using Leica LAS AF rel. 3.3 (Leica Microsystems GmbH) software. Images were processed using ImageJ and Adobe Photoshop software.

### 4.9. Cell Proliferation Assay of 2D and 3D Cultures

Mock/shRNASET2-OAW42 cells were seeded at 3000 cells/well in 96-well plates. At the end of each time point, mitochondrial activity was measured with CellTiter-Glo^®^ Luminescent Cell Viability Assay, accordingly to the manufacturer’s instructions (Promega, Madison, WI, USA). Mock/ tRNASET2-SKOV3 cells were seeded at 10,000 cells/well in 48-well plates. At the end of each time point, cells were trypsinized, resuspended in medium and immediately counted. We did not analyze mock and tRNASET2-SKOV3 cells’ mitochondrial activity, as described for OAW42, due to interference of GFP protein expressed from cells and the assay. For 3D culture, cells were grown in Algimatrix^TM^ (ThermoFisher Scientific) for 10 days and then recovered after Algimatrix^TM^ dissolving, according to the manufacturers’ protocols. Each well was divided form manual count of the spheres and for evaluation of cell viability with CellTiter-Glo^®^ Luminescent Cell Viability Assay (Promega). Cell viability assay was performed with PP2 (0, 2.5, 5, 10, 20, and 40 μM) up to 72 h. Mock/shRNASET2-OAW42 cells were seeded at 3000 cells/well in a 96-well plate; then cell viability was assessed with CellTiter-Glo^®^ Luminescent Cell Viability Assay. Mock/tRNASET2-SKOV3 cells were seeded at 25,000 cells/well in a 24-well plate; at 72 h cells were trypsinized, resuspended in medium, and immediately counted.

### 4.10. Soft Agar Assay

The soft agar assay was performed as described [23].

### 4.11. Growth and Invasion on Matrigel^®^

Cells were seeded on μ-Slide ibiTreat (ibidi GmbH, Martinsried, Germany), upon a coating of Matrigel^®^, and monitored for 7 days. Previously, membrane of mock and shRNASET2-OAW42 cells was labeled with PKH26 Red Fluorescent Cell Linker Kits (Sigma-Aldrich) according to the manufacturers’ protocols. Confocal microscopy was carried out using a Leica TCS SP8 X confocal laser scanning microscope (Leica Microsystems GmbH). Images were acquired in a z-size of 80.00 μm, with a step size of 2 μm, using a 20× objective. The Leica LAS AF rel. 3.3 (Leica Microsystems GmbH) software was used for 3D reconstruction and image analysis. This assay was then performed adding or not the src-inhibitor PP2 (Sigma-Aldrich) 10 μM.

### 4.12. FACS Analysis

Cells were incubated for 30 min with the primary antibody and 30 min with the secondary antibody, in ice. Cells were then fixed with PBS containing 1% of formaldehyde. Samples were analyzed using BD FACSCanto^TM^ and FlowJo software.

### 4.13. Adhesion Assay and Treatment with Inhibitors

After starvation, cells were trypsinized and maintained in suspension, in an incubator, for 1 h. Hence, cells were seeded on a coating of fibronectin 5 μg/mL (for mock/shRNASET2-OAW42) or collagen type I 5 μg/mL (for mock/tRNASET2-SKOV3), in presence or not of PP2 20 μM. After 30 min both adherent and suspended cells were lysed as described in WB session. Adhesion of mock/shRNASET2 OAW42 cells on collagen I or fibronectin was monitored up to 8 h, measuring cell confluence with JuLI™ Stage microscope and software (NanoEntek, South Corea).

### 4.14. Statistical Analysis

GraphPad Prism software (GraphPad Software, San Diego, CA, USA) was used to analyze all data. Differences between mean values were determined by Student′s *t*-test. Each experiment was performed at least three times for each condition; representative experiments are shown.

## 5. Conclusions

We provide the first direct evidence that RNASET2 expression, at least at the transcript level, which characterizes an EOC subgroup with better outcome. Furthermore, although RNASET2 has not yet been demonstrated to represent a therapeutic target ‘per se’, our findings argue for the notion that RNASET2-expressing EOC cells establish a peculiar interaction with ECM thus displaying reduced src activation. Therefore, the design of new therapeutic strategies with drugs targeting the SFK pathway appears to be worthy in RNASET2-expressing subgroup of EOCs.

## Figures and Tables

**Figure 1 cancers-11-00255-f001:**
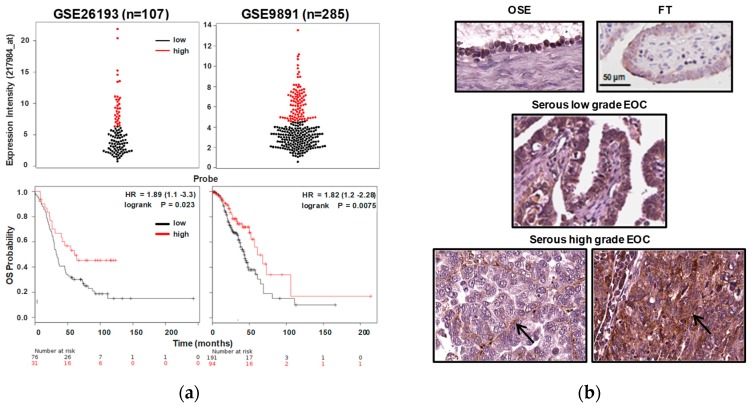
RNASET2 transcript expression in epithelial ovarian cancer (EOC) is associated with better prognosis. (**a**) Correlation of RNASET2 expression and overall survival (OS) was analyzed in GSE26193 (left panel) and GSE9891 (right panel) datasets. RNASET2 expression intensity is reported on the top, the Kaplan–Meyer plots are reported on the bottom. (**b**) Representative images of immunohistochemistry (IHC) with anti-RNASET2 Ab on normal ovarian (OSE) and fallopian tube (FT) epithelia, and on representative serous low grade and high grade EOC samples, as reported in Table 1. Arrows highlight RNASET2 staining at the levels of extracellular matrix (ECM) deposition.

**Figure 2 cancers-11-00255-f002:**
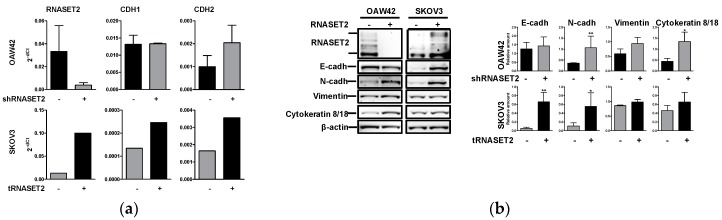
RNASET2 depletion causes phenotypic changes in EOC cellular models. (**a**) Real-time RT-PCR showing the transcript levels of RNASET2, CDH1, and CDH2 in mock/shRNASET2-OAW42 and mock/tRNASET2-SKOV3 cells. Results are presented as relative expression normalized to GAPDH mRNA levels. Error bars, SD. (**b**) Left panel: Western blotting on lysates from mock/shRNASET2-OAW42 and mock/tRNASET2-SKOV3 cells. Immunoblottings were performed with Abs against the proteins reported on the left. β-actin was used as control for gel loading. Right panel: quantitative evaluation of the proteins in three experiments. The graph reports the ratio between the target protein and β-actin. Asterisks indicate statistically significant values by Student’s *t*-test (* *p* ˂ 0.05; ** *p* ˂ 0.01). (**c**) Immunofluorescence (IF) analysis on fixed mock/shRNASET2-OAW42 and mock/tRNASET2-SKOV3 cells performed with Abs reported on the left.

**Figure 3 cancers-11-00255-f003:**
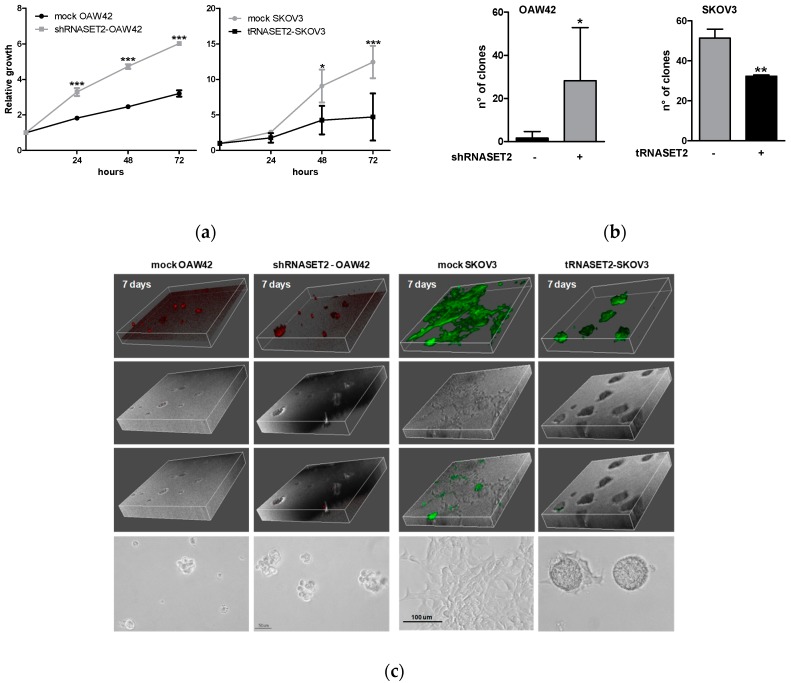
RNASET2 expression negatively impinges EOC cell growth. (**a**) Left panel: proliferation assay performed on mock and shRNASET2-OAW42 cells. Cell viability was measured by CellTiter-Glo^®^ Luminescent Cell viability assay (Promega). Results are presented as relative growth to T0. Asterisks indicate significant values (*** *p* ˂ 0.001). Right panel: proliferation assay performed on mock and tRNASET2-SKOV3 cells. Number of cells was determined by a manual count. Results are presented as relative growth to T0. Asterisk indicates significant values (* *p* ˂ 0.05 and *** *p* ˂ 0.001). Error bars, SD. (**b**) Clonogenic assay of mock/shRNASET2-OAW42 (left panel) and mock/tRNASET2-SKOV3 cells (right panel) grown in soft agar for 20 days. The graphs report the number of clones/well. Asterisk indicates significant values (* *p* ˂ 0.05 and ** *p* < 0.01). Error bars, SD. (**c**) Three-dimensional deconvolution of confocal images of cells growing and invading Matrigel^®^ (Corning, New York, NY, USA) at 7 days. From top to bottom: fluorescent images, dichroic images, merge of both and phase contrast images of the same cells. Left panel: mock and shRNASET2-OAW42 cells labeled with PKH26 Red Fluorescent Cell Linker Kits (Sigma-Aldrich) as described in Materials and Methods section. Right panel: mock and tRNASET2-SKOV3 cells stably transfected with a pcDNA3-based expression vector encoding GFP alone or RNASET2 and GFP, respectively.

**Figure 4 cancers-11-00255-f004:**
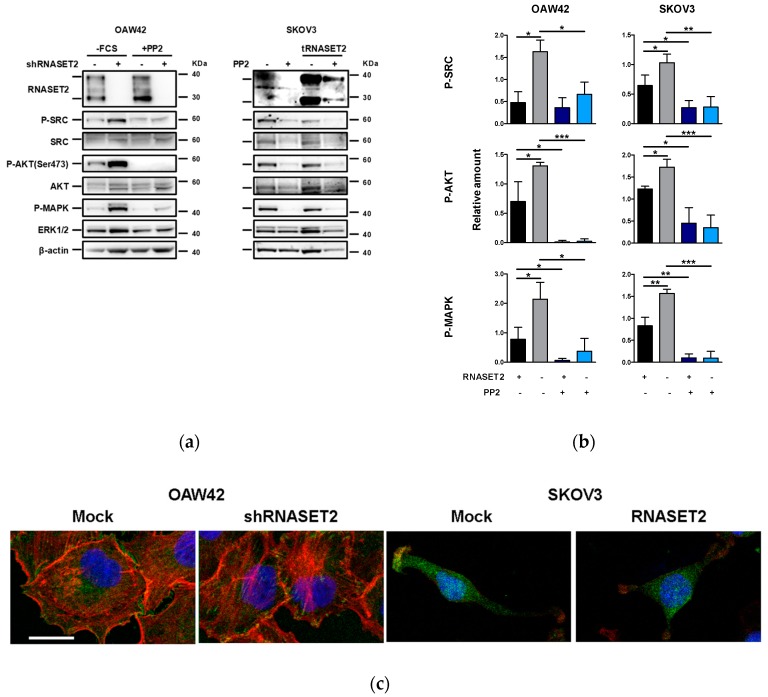
‘RNASET2 phenotype’ negatively modulates src activation. (**a**) Western blotting on lysates from EOC cells untreated or treated with PP2 (20 μM) and then left to adhere on fibronectin- coated (5 μg/mL, for OAW42 cells) or on collagen type I- coated (5 μg/mL, for SKOV3 cells) wells for 30 min. A representative experiment with each cell line (OAW42 or SKOV3) of three performed is shown. (**b**) Quantitative evaluation of P-SRC, P-AKT and P-MAPK analyzed in the Western blotting (WB) of three different experiments. The graph reports the ratio between the target protein and β-actin. Asterisks indicate statistically significant values by Student’s *t*-test (* *p* ˂ 0.05; ** *p* ˂ 0.01; *** *p* < 0.001). Immunoblottings were performed with Abs against the proteins reported on the left. β-actin was used as control for gel loading. Dark and light blue boxes highlight cell treated with PP2. (**c**) Confocal IF with antiphosphorylated paxillin (green) and phalloidin (red) performed on transfected EOC cells after adhesion to fibronectin (OAW42) or collagen I (SKOV3). Nuclei were stained with Dapi. Bar, 20 µm. Larger captured images are reported in Appendix A.

**Figure 5 cancers-11-00255-f005:**
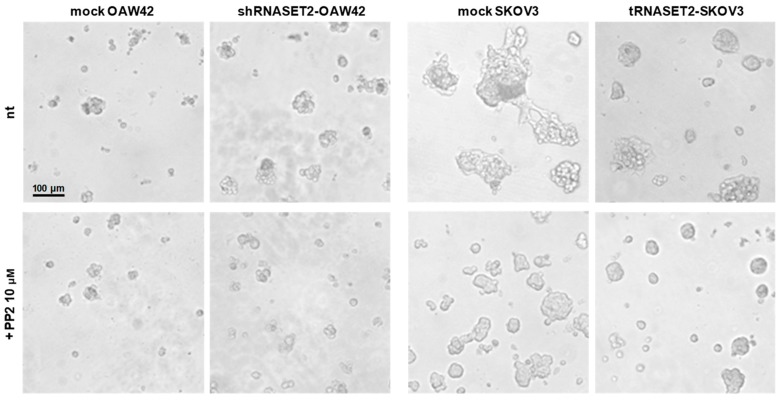
Src inhibitor restores ‘RNASET2 phenotype’. Representative phase contrast images of mock- or shRNASET2-OAW42 (left panel) and mock- or tRNASET2-SKOV3 cells (right panel) growing and invading Matrigel^®^ up to 7 days with or without PP2 (10 μM).

**Figure 6 cancers-11-00255-f006:**
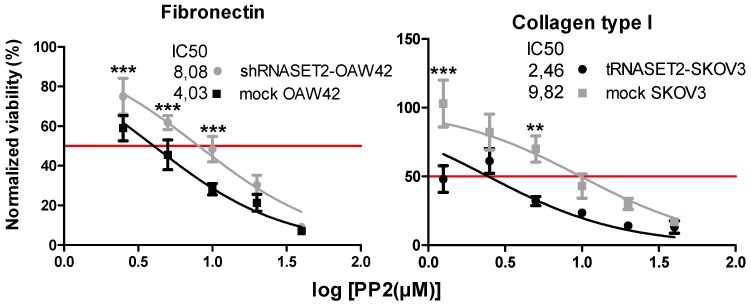
EOC cells with the ‘RNASET2 phenotype’ are sensitive to src inhibitor. Cell viability assay of mock/shRNASET2-OAW42 (left panel) and mock/tRNASET2-SKOV3 cells (right panel) treated with PP2 (0, 2.5, 5, 10, 20, and 40 μM) up to 72 h. Asterisk indicates significant values (*** *p* < 0.001 and ** *p* < 0.01). Error bars, SD.

**Table 1 cancers-11-00255-t001:** Immunohistochemical analysis with anti-RNASET2 Ab on formalin-fixed paraffin-embedded EOC tissue sections.

Histological Subtype		RNASET2 Expression		
		Negative (27.5%)	Positive (72.5%)	
		Score ** 0 (*n* = 27)	Score 1 (*n* = 43)	Score 2 (*n* = 32)
OSE *		-	-	1
FTE *		-	1	-
Type I (*n* = 47)	Endometrioid	0	2	2
	Serous	4	10	3
	Mucinous	4	9	7
	Clear cell	0	3	3
Type II (*n* = 54)	Endometrioid	2	4	0
	Serous	13	16	16
	Undifferentiated	3	0	0

***** OSE, ovarian surface epithelium; FTE, fallopian tube epithelium. ****** Score: 0, negative; 1, moderately staining intensity only in the tumor-associated ECM; 2, strong staining intensity both in the cytoplasm and in the tumor-associated ECM.

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
