# Peer review of "A Cell-Autonomous Oncosuppressive Role of Human RNASET2 Affecting ECM-Mediated Oncogenic Signaling"

_cancers, 2019, doi:10.3390/cancers11020255_

Round 1
Reviewer 1 Report
This is a well written manuscript in which authors have investigated the role of RNASET2 in ECM-mediated progression of EOCs. The presented data supports some of the conclusions presented in the manuscript, however there are areas in which the data is not strong enough to support the arguments/conclusions made by the authors. Following are the specific comments which could further enhance the quality of this manuscript.
The IHC data is not strong enough to support the conclusions. a) A total of 75 samples (43 with score 1 and 32 with score 2) have low to high levels of RNASET2, and only 27 were negative. Therefore, how does the protein levels of RNASET2 are correlated with EOC progression? this is not clear. b) What was the score for the staining of RNASET2 in OSE and FT? It would be great to include the staining score for OSE and FT in the table so that the readers could compare it with the cancer samples.
SKOV3 may not be a good representative model for ovarian cancer (Domke et at. Nat. Comm.). did authors test their hypothesis on any other ovarian cancer cell line? Also, fig 2b shows that skov3 express low levels of RNASET2. how much of RNASET2 is enough to give a phenotype? is the low levels of RNASET2 in skov3 are representative of IHC samples with score 1?
The IF images are not high quality (especially for OAW42) to clearly identify focal adhesions or the stress fibers. Could authors take better images? Also, it would have been useful to include Vimentin staining to look at the focal adhesions because the authors talk about focal adhesions. Are these confocal images? The present IF data is weak to support the authors' arguments.
Did the authors look at the extracellular release of RNASET2 in both cell lines? It could be done by testing the conditioned media for RNASET2 using WB.
Line 230: Why the data is not shown? Please include the data in the supplementary.
Since the authors talk about the role of RNASET2 in EM interation and modulation, did the authors try trichrome staining for the ECM in tissue samples with low vs high RNASET2 levels?
Author Response
Reviewer 1
Rebuttal to Reviever’s comment at the Manuscript ID Cancers-441624
On behalf of all authors we thank this reviewer for her/his valuable comments which helped us to prepare a revised version of the manuscript. Following comments/suggestion made by the Reviewers the manuscript has been carefully revised and the text edited, especially the Discussion section. The changes made in the text are highlighted in red.
Our point-by-point response to each comment is given below.
This is a well written manuscript in which authors have investigated the role of RNASET2 in ECM-mediated progression of EOCs. The presented data supports some of the conclusions presented in the manuscript, however there are areas in which the data is not strong enough to support the arguments/conclusions made by the authors. Following are the specific comments which could further enhance the quality of this manuscript.
The Authors thanks this Reviewer for this general favourable comment.
The IHC data is not strong enough to support the conclusions. a) A total of 75 samples (43 with score 1 and 32 with score 2) have low to high levels of RNASET2, and only 27 were negative. Therefore, how does the protein levels of RNASET2 are correlated with EOC progression? this is not clear. b) What was the score for the staining of RNASET2 in OSE and FT? It would be great to include the staining score for OSE and FT in the table so that the readers could compare it with the cancer samples.
The text describing these data has been extensively edited to avoid misinterpretations since a correlation with clinical endpoint was not possible for this patients’ cohort. However, among samples staining positive for RTNASET2 expression, only 32% showed a high expression in agreement with transcript expression data. As suggested, the expression intensity of RNASET2 in ovarian and tubal epithelia has been included in Table 1.
SKOV3 may not be a good representative model for ovarian cancer (Domke et at. Nat. Comm.). did authors test their hypothesis on any other ovarian cancer cell line? Also, fig 2b shows that skov3 express low levels of RNASET2. how much of RNASET2 is enough to give a phenotype? is the low levels of RNASET2 in skov3 are representative of IHC samples with score 1?
We have previously reported data obtained using the OVCAR3 cell line which constitutively expresses RNASET2 and indeed RNASET2 was silenced in these cells to reveal its role (see References 8,9,14). We are aware about the controversy concerning the SKOV3 cell line, usually used by different laboratories worldwide. Indeed, the p53 status of this cell line, reported as wild type in the Nature Comm paper, has been extensively re-evaluated by subsequent reports in which the cell line has been defined as null for p53 (Beaufort, et al. Ovarian cancer cell line panel (OCCP): clinical importance of in vitro morphological subtypes. PLoS One 2014, 9, e103988; Hernandez, L.; et al. Characterization of ovarian cancer cell lines as in vivo models for preclinical studies. Gynecol Oncol 2016, 142, 332-340). We have now cited all three papers as references 18-20 and explained in the text our choices (lanes....). In this particular study, we wanted to investigate whether RNASET2 could behave differently in cells with different grade of differentiation. Therefore, we have chosen SKOV3 as less differentiated RNASET2-low expressing EOC cell line with a more mesenchymal morphology, and OAW42 well differentiated RNASET2 expressing cells as representative of epithelial EOC cells. These aspects have been clarified in the text (lanes 117-121)
SKOV3 cells appear to express low levels of RNASET2 although a direct comparison with IHC samples with score 1 is not possible due to the different methodology used for protein detection. ‘A clear cut-off of RNASET2 expression for phenotype definition is not yet available, however our data indicate that the expression of RNASET2 detected in SKOV3 is not sufficient to exert an anti-tumor activity which is obtained only after protein over-expression’. This concept has been now added in the Discussion section, lanes 352-354.
We thank the Reviewer for this particular comment.
The IF images are not high quality (especially for OAW42) to clearly identify focal adhesions or the stress fibers. Could authors take better images? Also, it would have been useful to include Vimentin staining to look at the focal adhesions because the authors talk about focal adhesions. Are these confocal images? The present IF data is weak to support the authors' arguments.
The IF images were substituted accordingly to these comments and we also added in Fig. 4 and S5 the analysis of focal adhesions stained for phosphorylated paxillin.
Did the authors look at the extracellular release of RNASET2 in both cell lines? It could be done by testing the conditioned media for RNASET2 using WB.
We thank the reviewer for this suggestion; we have now included these data in Fig. S4.
Line 230: Why the data is not shown? Please include the data in the supplementary.
We have included these data in Fig. S3b.
Since the authors talk about the role of RNASET2 in EM interaction and modulation, did the authors try trichrome staining for the ECM in tissue samples with low vs high RNASET2 levels?
This data were already reported by us (Acquati et al. PNAS 2011) on tumor samples from mouse xenografts and other images are reported below. Indeed, RNASET2 expression resulted to be associated to a peculiar collagen organization, as we have now reported in Discussion section, lanes 360-362. Analysis of ECM, evaluated with different approaches, using human EOC samples, is the object of a different project and the data will be collected and reported elsewhere. However, up to the Reviewer, we can add these pictures in the present version of the manuscript as supplementary figure.
We would like to thank again the Reviewer for her/his fruitful suggestions/comments that helped us to improve the quality of our manuscript.
We hope that the present version of the manuscript is now suitable for publication in Cancers.
Antonella Tomassetti and Francesco Acquati
Corresponding Authors

Reviewer 2 Report
In this manuscript, Francesca et al. investigated the effects of RNASET2 expression modulation oc cell phenotype and behavior in EOC cell models. Their results showed that RNASET2 transcript expression in EOC is associated to better prognosis. By using two cell models with overexpressed or silenced RNASET2, the authors found that RNASET2 expression negatively affects growth capability by conferring a peculiar cell phenotype upon the interaction of EOC cells with the ECM, which mainly due to decreased src kinase activation. Their results suggest that activated src could be used as a therapeutic target in the treatment of RNASET2-expressing EOCs. However, there are several concerns the authors will want to address to publish an interpretable story for the reader.
The different subtypes of ovarian cancer are distinct diseases that contain different gene mutation profiles which may affect src activation. Could the author provide information about this aspect?
In some figures, there is no asterisks to indicate significant values.
How many independent experiments backed up the ones shown in Fig2b? In addition to that, the quantitative data only evaluated one time result. The author should provide statistical results for the western blotting. The same goes to fig4.
The specific inhibitor of src was used to explore its involvement and function, however, the siRNA technology should be applied to these experiment to further confirm the inhibitor results.
It would be better if the authors could provide the results about inhibition of src activation in RNASET-2 expressing EOCs in in vivo xenograft-based animal models.
Line 43 “…last 30 years due intrinsic…” it should be “due to”. Line 82 “…including ovarian” it should be “ovarian cancer”. Line 230 “…SKOV3 cell..” It should be “cells”.
Author Response
Reviewer 2
Rebuttal to Reviever’s comment at the Manuscript ID Cancers-441624
On behalf of all authors we thank this reviewer for her/his valuable comments which helped us to prepare a revised version of the manuscript. Following comments/suggestion made by the Reviewers the manuscript has been carefully revised and the text edited, especially the Discussion section. The changes made in the text are highlighted in red.
Our point-by-point response to each comment is given below.
Comments and Suggestions for Authors
In this manuscript, Francesca et al. investigated the effects of RNASET2 expression modulation oc cell phenotype and behavior in EOC cell models. Their results showed that RNASET2 transcript expression in EOC is associated to better prognosis. By using two cell models with overexpressed or silenced RNASET2, the authors found that RNASET2 expression negatively affects growth capability by conferring a peculiar cell phenotype upon the interaction of EOC cells with the ECM, which mainly due to decreased src kinase activation. Their results suggest that activated src could be used as a therapeutic target in the treatment of RNASET2-expressing EOCs. However, there are several concerns the authors will want to address to publish an interpretable story for the reader.
The different subtypes of ovarian cancer are distinct diseases that contain different gene mutation profiles which may affect src activation. Could the author provide information about this aspect?
No data are reported on SFK activations in the different EOC histotype or related to the mutations specifically present in the different histotype (i.e. p53 mutation of serous EOC respect to src activations). A sentence clarifying this issue has been added in the Discussion section of the revised manuscript (lanes 373-374).
In some figures, there is no asterisks to indicate significant values.
How many independent experiments backed up the ones shown in Fig2b? In addition to that, the quantitative data only evaluated one time result. The author should provide statistical results for the western blotting. The same goes to fig4.
Asterisks referred to significant values have been accordingly added to figures. We now show quantitations followed by statistical analyses performed on three or more experiments of western blotting.
The specific inhibitor of src was used to explore its involvement and function, however, the siRNA technology should be applied to these experiment to further confirm the inhibitor results.
We thank the reviewer for this comment and indeed PP2 is highly effective in inhibiting all src family kinases (SFKs) and therefore the text has been modified accordingly adding a panel in Fig. 4.
It would be better if the authors could provide the results about inhibition of src activation in RNASET-2 expressing EOCs in in vivo xenograft-based animal models.
As said in Discussion section, lane 382-383, we are developing a different project for drug susceptibility selecting tumors from PDXs expressing or not expressing RNASET2 and the data will be ready later on.
Line 43 “…last 30 years due intrinsic…” it should be “due to”. Line 82 “…including ovarian” it should be “ovarian cancer”. Line 230 “…SKOV3 cell..” It should be “cells”.
We apologize for these typing mistakes. The changes have been made in the new version of the manuscript.
We would like to thank again the Reviewer for her/his fruitful suggestions/comments that helped us to improve the quality of our manuscript.
We hope that the present version of the manuscript is now suitable for publication in Cancers.
Antonella Tomassetti and Francesco Acquati
Corresponding Authors
